# Text2NKG: Fine-Grained N-ary Relation Extraction for N-ary relational Knowledge Graph Construction

Haoran Luo[1], Haihong E[1]*, Yuhao Yang[2], Tianyu Yao[1],
Yikai Guo[3], Zichen Tang[1], Wentai Zhang[1], Shiyao Peng[1], Kaiyang Wan[1],
Meina Song[1], Wei Lin[4], Yifan Zhu[1], Luu Anh Tuan[5]

[1]School of Computer Science, Beijing University of Posts and Telecommunications, China
[2]School of Automation Science and Electrical Engineering, Beihang University, China
[3]Beijing Institute of Computer Technology and Application [4]Inspur Group Co., Ltd., China
[5]College of Computing and Data Science, Nanyang Technological University, Singapore
{luohaoran, ehaihong, yifan_zhu}@bupt.edu.cn, anhtuan.luu@ntu.edu.sg

## Abstract

Beyond traditional binary relational facts, n-ary relational knowledge graphs (NKGs) are comprised of n-ary relational facts containing more than two entities, which are closer to real-world facts with broader applications. However, the construction of NKGs remains at a coarse-grained level, which is always in a single schema, ignoring the order and variable arity of entities. To address these restrictions, we propose Text2NKG, a novel fine-grained n-ary relation extraction framework for n-ary relational knowledge graph construction. We introduce a span-tuple classification approach with hetero-ordered merging and output merging to accomplish fine-grained n-ary relation extraction in different arity. Furthermore, Text2NKG supports four typical NKG schemas: *hyper-relational schema*, *event-based schema*, *role-based schema*, and *hypergraph-based schema*, with high flexibility and practicality. The experimental results demonstrate that Text2NKG achieves state-of-the-art performance in $F_1$ scores on the fine-grained n-ary relation extraction benchmark. Our code and datasets are publicly available[1].

## 1 Introduction

Modern knowledge graphs (KGs), such as Freebase [2], Google Knowledge Vault [7], and Wikidata [21], utilize a multi-relational graph structure to represent knowledge. Because of the advantage of intuitiveness and interpretability, KGs find various applications in question answering [28], query response [1], logical reasoning [4], and recommendation systems [29]. Traditional KGs are mostly composed of binary relational facts ($subject$, $relation$, $object$), which represent the relationship between two entities [3]. However, it has been observed [20] that over 30% of real-world facts involve n-ary

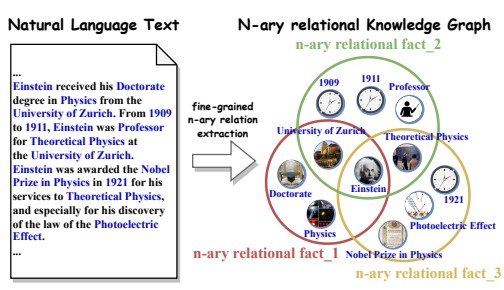

Figure 1: An example of NKG construction.

relation facts with more than two entities ($n \geq 2$). As shown in Figure 1, an n-ary relational knowledge graph (NKG) is composed of many n-ary relation facts, offering richer knowledge expression

---

* Corresponding author.

[1] https://github.com/LHRLAB/Text2NKG

38th Conference on Neural Information Processing Systems (NeurIPS 2024).

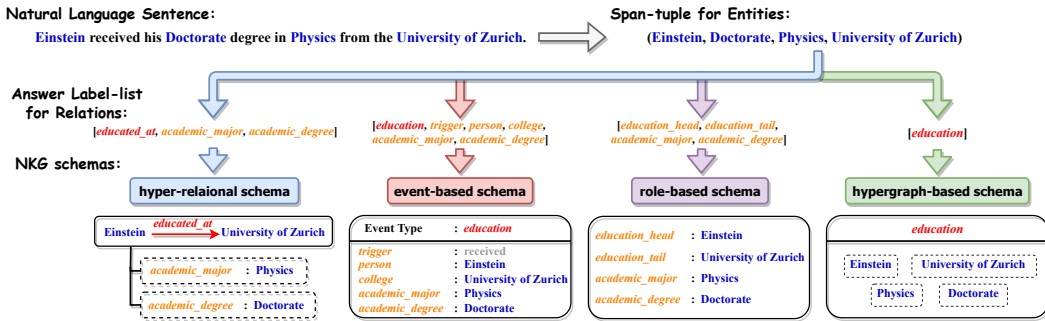

Figure 2: Taking a real-world textual fact as an example, we can extract a four-arity structured span-tuple for entities (`Einstein, University of Zurich, Doctorate, Physics`) with an answer label-list for relations accordingly as a 4-ary relational fact from the sentence through n-ary relation extraction.

and wider application capabilities. As a key step of constructing NKGs, n-ary relation extraction (n-ary RE) is a task of identifying n-ary relations among entities in natural language texts. Compared to binary relational facts, n-ary relational facts in NKGs have more diverse schemas for different scenarios. For example, Wikidata utilizes n-ary relational facts in a *hyper-relational schema* [20, 10, 23], i.e., $(s, r, o, \{(k_i, v_i)\}_{i=1}^{n-2})$ which adds $(n-2)$ key-value pairs to the main triple to represent auxiliary information. In addition to the *hyper-relational schema*, the existing NKG schemas also include *event-based schema* $(r, \{(k_i, v_i)\}_{i=1}^{n})$ [11, 16], *role-based schema* $(\{(k_i, v_i)\}_{i=1}^{n})$ [12, 15], and *hypergraph-based schema* $(r, \{v_i\}_{i=1}^{n})$ [26, 8], as shown in Figure 2.

Currently, most existing NKGs in four schemas, such as JF17K [26], Wikipeople [12], WD50K [10], and EventKG [11], are manually constructed. Previous n-ary RE methods [13, 31] focus on extraction with a fixed number of entities in *hypergraph-based schema* or *role-based schema*. Existing event extraction methods [16, 17, 9] can achieve n-ary RE in *event-based schema*. Recently, CubeRE [5] introduce a cube-filling method, which is the only n-ary RE method in *hyper-relational schema*.

However, there are still three main challenges in automated n-ary RE for NKG construction, which remains at a coarse-grained level: **(1) Diversity of NKG schemas.** Previous methods could only perform N-ary RE based on a specific schema, but currently, there is no flexible method that can perform n-ary RE for arbitrary schema with different number of relations. **(2) Determination of the order of entities.** N-ary RE involves more possible entity orders than binary RE, for example, as shown in Figure 2, in a *hyper-relational schema*, there is an order issue regarding which entity is the head entity, tail entity, or auxiliary entity. Previous methods often ignored the joint impact of different entity orders, leading to inaccurate extraction.**(3) Variability of the arity of n-ary RE.** Previous methods usually output a fixed number of entities and are not adept at determining the variable number of entities forming an n-ary relational fact.

To tackle these challenges, we introduce **Text2NKG**, a novel fine-grained n-ary RE framework designed to automate the generation of n-ary relational facts from natural language text for NKG construction. Text2NKG employs a **span-tuple multi-label classification** method, which transforms n-ary RE into a multi-label classification task for span-tuples, including all combinations of entities in the text. Because the number of predicted relation labels corresponds to the chosen NKG schema, Text2NKG is adaptable to all NKG schemas, offering examples with *hyper-relational schema*, *event-based schema*, *role-based schema*, and *hypergraph-based schema*, all of which have broad applications. Moreover, Text2NKG introduces a **hetero-ordered merging** method, considering the probabilities of predicted labels for different entity orders to determine the final entity order. Finally, Text2NKG proposes an **output merging** method, which is used to unsupervisedly derive n-ary relational facts of any number of entities for NKG construction.

In addition, we extend the only n-ary RE benchmark for NKG construction, HyperRED [5], which is in the *hyper-relational schema*, to four NKG schemas. We've done sufficient n-ary RE experiments on HyperRED, and the experimental results show that Text2NKG achieves state-of-the-art performance in $F_1$ scores of hyper-relational extraction. We also compared the results of Text2NKG in the other three schemas to verify applications.

## 2   Related Work

### 2.1   N-ary relational Knowledge Graph

An n-ary relational knowledge graph (NKG) consists of n-ary relational facts, which contain $n$ entities ($n \geq 2$) and several relations. The n-ary relational facts are necessary and cannot be replaced by combinations of some binary relational facts because we cannot distinguish which binary relations are combined to represent the n-ary relational fact in the whole KG. Therefore, NKG utilizes a schema in every n-ary relational fact locally and a hypergraph representation globally [18].

Firstly, the simplest NKG schema is hypergraph-based. [26] found that over 30% of Freebase [2] entities participate facts with more than two entities, first defined n-ary relations mathematically and used star-to-clique conversion to convert triple-based facts representing n-ary relational facts into the first NKG dataset JF17K in *hypergraph-based schema* $(r, \{v_i\}_{i=1}^n)$. [8] proposed FB-AUTO and M-FB15K with the same *hypergraph-based schema*. Secondly, [12] introduced role information for n-ary relational facts and extracted Wikipeople, the first NKG dataset in *role-based schema* $(\{(k_i, v_i)\}_{i=1}^n)$, composed of role-value pairs. Thirdly, Wikidata [21], the largest knowledge base, utilizes an NKG schema based on hyper-relation $(s, r, o, \{(k_i, v_i)\}_{i=1}^{n-2})$, which adds auxiliary key-value pairs to the main triple. [10] first proposed an NKG dataset in *hyper-relational schema* WD50K. Fourthly, as [11] pointed out, events are also n-ary relational facts. One basic event representation has an event type, a trigger, and several key-value pairs [16]. Regarding the event type as the main relation, the (trigger: value) as one of the key-value pairs, and the arguments as the rest key-value pairs, we can obtain an *event-based NKG schema* $(r, \{(k_i, v_i)\}_{i=1}^n)$.

Based on four common NKG schemas, we propose Text2NKG, the first method for extraction of structured n-ary relational facts from natural language text, which improves NKG representation and application.

### 2.2   N-ary Relation Extraction

Relation extraction (RE) is an important step of KG construction, directly affecting the quality, scale, and application of KGs. While most of the current n-ary relation extraction (n-ary RE) for NKG construction depends on manual construction [26, 12, 10] but not automated methods. Most automated RE methods target the extraction of traditional binary relational facts. For example, [22] proposes a table-filling method for binary RE, and [30, 27] propose span-based RE methods with levitated marker and packed levitated marker, respectively.

For automated n-ary RE, some approaches [13, 31] treat n-ary RE in *hypergraph-based schema* or *role-based schema* as a binary classification problem and predict whether the composition of n-ary information in a document is valid or not. However, these methods extract n-ary information in fixed arity, which are not flexible. Moreover, some event extraction methods [16, 17, 9] propose different event trigger and argument extraction techniques, which can achieve n-ary RE in *event-based schema*. Recently, CubeRE [5] proposes an automated n-ary RE method in *hyper-relational schema*, which extends the table-filling extraction method to n-ary RE with cube-filling. However, these methods can only model one of the useful NKG schemas with limited extraction accuracy.

In this paper, we propose the first fine-grained n-ary RE framework Text2NKG for NKG construction in four example schemas, proposing a span-tuple multi-label classification method with hetero-ordered merging and output merging to improve the accuracy of fine-grained n-ary RE extraction in all NKG schemas substantially.

## 3   Preliminaries

**Formulation of NKG.** An NKG $\mathcal{G} = \{\mathcal{E}, \mathcal{R}, \mathcal{F}\}$ consists of an entity set $\mathcal{E}$, a relation set $\mathcal{R}$, and an n-ary fact ($n \geq 2$) set $\mathcal{F}$. Each n-ary fact $f^n \in \mathcal{F}$ consists of entities $\in \mathcal{E}$ and relations $\in \mathcal{R}$. For hyper-relational schema [20]: $f_{hr}^n = (e_1, r_1, e_2, \{r_{i-1}, e_i\}_{i=3}^n)$ where $\{e_i\}_{i=1}^n \in \mathcal{E}$, $\{r_i\}_{i=1}^{n-1} \in \mathcal{R}$. For event-based schema [16]: $f_{ev}^n = (r_1, \{r_{i+1}, e_i\}_{i=1}^n)$, where $\{e_i\}_{i=1}^n \in \mathcal{E}$, $\{r_i\}_{i=1}^{n+1} \in \mathcal{R}$. For role-based schema [12]: $f_{ro}^n = (\{r_i, e_i\}_{i=1}^n)$, where $\{e_i\}_{i=1}^n \in \mathcal{E}$, $\{r_i\}_{i=1}^n \in \mathcal{R}$. For hypergraph-based schema [26]: $f_{hg}^n = (r_1, \{e_i\}_{i=1}^n)$, where $\{e_i\}_{i=1}^n \in \mathcal{E}$, $r_1 \in \mathcal{R}$.

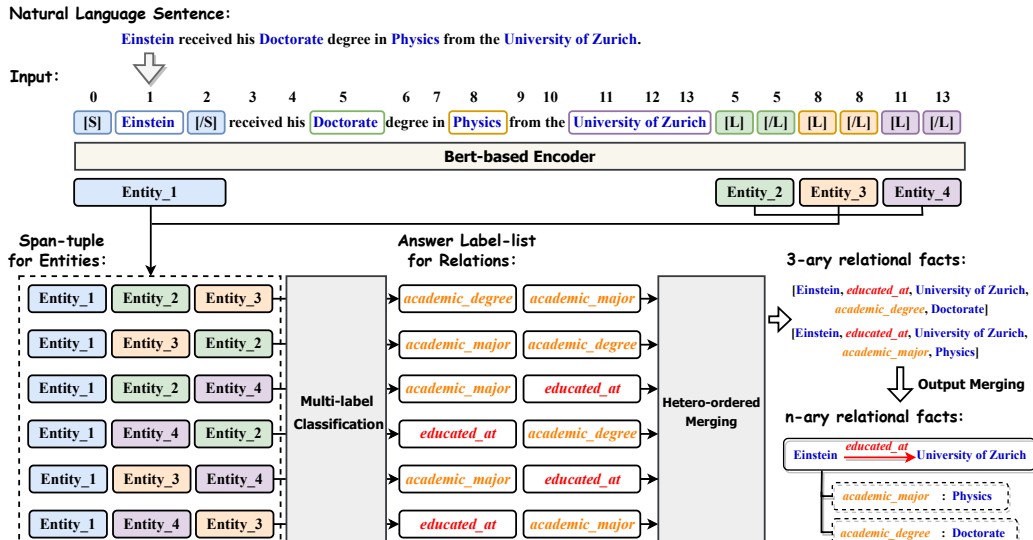

Figure 3: An overview of Text2NKG extracting n-ary relation facts from a natural language sentence in hyper-relational NKG schema for an example.

**Problem Definition.** Given an input sentence with $l$ words $s = \{w_1, w_2, ..., w_l\}$, an entity $e$ is a consecutive span of words: $e = \{w_p, w_{p+1}, ..., w_q\} \in \mathcal{E}_s$, where $p, q \in \{1, ..., l\}$, and $\mathcal{E}_s = \{e_j\}_{j=1}^m$ is the entity set of all $m$ entities in the sentence. The output of n-ary relation extraction, $R()$, is a set of n-ary relational facts $\mathcal{F}_s$ in given NKG schema in $\{f_{hr}^n, f_{ev}^n, f_{ro}^n, f_{hg}^n\}$. Specifically, each n-ary relational fact $f^n \in \mathcal{F}_s$ is extracted by multi-label classification of one of the ordered span-tuple for $n$ entities $[e_i]_{i=1}^n \in \mathcal{E}_s$, forming an answer label-list for $n_r$ relations $[r_i]_{i=1}^{n_r} \in \mathcal{R}$, where $n$ is the arity of the extracted n-ary relational fact, and $n_r$ is the number of answer relations in the fact, which is determined by the given NKG schema: $R([e_i]_{i=1}^n) = [r_i]_{i=1}^{n-1}$, when $f^n = f_{hr}^n$, $R([e_i]_{i=1}^n) = [r_i]_{i=1}^{n+1}$ when $f^n = f_{ev}^n$, $R([e_i]_{i=1}^n) = [r_i]_{i=1}^n$ when $f^n = f_{ro}^n$, and $R([e_i]_{i=1}^n) = [r_1]$ when $f^n = f_{hg}^n$.

## 4 Methodology

In this section, we first introduce the overview of the Text2NKG framework, followed by the span-tuple multi-label classification, training strategy, hetero-ordered merging, and output merging.

### 4.1 Overview of Text2NKG

Text2NKG is a fine-grained n-ary relation extraction framework built for n-ary relational knowledge graph (NKG) construction. The input to Text2NKG is natural language text tokens labeled with entity span in sentence units. First, inspired by [27], Text2NKG encodes the entities using BERT-based Encoder [6] with a packaged levitated marker for embedding. Then each arrangement of ordered span-tuple with three entity embeddings will be classified with multiple labels, and the framework will be learned by the weighted cross-entropy with a null-label bias. In the decoding stage, in order to filter the n-ary relational facts whose entity compositions have isomorphic hetero-ordered characteristics, Text2NKG proposes a hetero-ordered merging strategy to merge the label probabilities of $3! = 6$ arrangement cases of span-tuples composed of the same entities and filter out the output 3-ary relational facts existing non-conforming relations. Finally, Text2NKG combines the output 3-ary relational facts to form the final n-ary relational facts with output merging.

### 4.2 Span-tuple Multi-label Classification

For the given sentence token $s = \{w_1, w_2, ..., w_l\}$ and the set of entities $\mathcal{E}_s$, in order to perform fine-grained n-ary RE, we need first to encode a span-tuple $(e_1, e_2, e_3)$ consisting of every arrangement of three ordered entities, where $e_1, e_2, e_3 \in \mathcal{E}_s$. Due to the high time complexity of training every

span-tuple as one training item, inspired by [27], we achieve the reduction of training items by using packed levitated markers that pack one training item with each entity in $\mathcal{E}_s$ separately. Specifically, in each packed training item, a pair of solid tokens, [S] and [/S], are added before and after the packed entity $e_S = \{w_{p_S}, ..., w_{q_S}\}$, and $(|\mathcal{E}_s| - 1)$ pairs of levitated markers, [L] and [/L], according to other entities in $\mathcal{E}_s$, are added with the same position embeddings as the beginning and end of their corresponding entities span $e_{L_i} = \{w_{p_{L_i}}, ..., w_{q_{L_i}}\}$ to form the input token $\mathbf{X}$:

$$
\begin{aligned}
\mathbf{X} = \{ & w_1, ..., [S], w_{p_S}, ..., w_{q_S}, [/S], ..., \\
& w_{p_{L_i}} \cup [L], ..., w_{q_{L_i}} \cup [/L], ..., w_l \}.
\end{aligned}
\tag{1}
$$

We encode such token by the BERT-based pre-trained model encoder [6]:

$$
\{h_1, h_2, ..., h_t\} = \text{BERT}(\mathbf{X}),
\tag{2}
$$

where $t = |\mathbf{X}|$ is the input token length, $\{h_i\}_{i=1}^t \in \mathbb{R}^d$, and $d$ is embedding size.

There are several span-tuples $(A, B, C)$ in a training item. The embedding of first entity $h_A \in \mathbb{R}^{2d}$ in the span-tuple is obtained by concat embedding of the solid markers, [S] and [/S], and the embeddings of second and third entities $h_B, h_C \in \mathbb{R}^{2d}$ are obtained by concat embeddings of levitated markers, [L] and [/L] with all $A_{m-1}^2$ arrangement of any other two entities in $\mathcal{E}_s$. Thus, we obtain the embedding representation of the three entities to form $A_{m-1}^2$ span-tuples in one training item. Therefore, every input sentence contains $m$ training items with $m A_{m-1}^2 = A_m^3$ span-tuples for any ordered arrangement of three entities.

We then define $n_r$ linear classifiers, each of which consists of 3 feedforward neural networks $\{\text{FNN}_i^k\}_{i=1}^{n_r}, k = 1, 2, 3$, to classify the span-tuples for multiple-label classification. Each classifier targets the prediction of one relation $r_i$, thus obtaining a probability lists $(\mathbf{P}_i)_{i=1}^{n_r}$ with all relations in given relation set $\mathcal{R}$ plus a null-label:

$$
\mathbf{P}_i = \text{FNN}_i^1(h_A) + \text{FNN}_i^2(h_B) + \text{FNN}_i^3(h_C),
\tag{3}
$$

where $\text{FNN}_i^k \in \mathbb{R}^{2d \times (|\mathcal{R}|+1)}$, and $\mathbf{P}_i \in \mathbb{R}^{(|\mathcal{R}|+1)}$.

### 4.3 Training Strategy

To train the $n_r$ classifiers for each relation prediction more accurately, we propose a data augmentation strategy for span-tuples. Taking the *hyper-relational schema* as an example, given a hyper-relational fact $(A, r_1, B, r_2, C)$, we consider swapping the head and tail entities, and changing the main relation to its inverse $(B, r_1^{-1}, A, r_2, C)$, as well as swapping the tail entities with auxiliary values, and the main relation with the auxiliary key $(A, r_2, C, r_1, B)$, also as labeled training span-tuple cases. Thus $R_{hr}(A, B, C) = (r_1, r_2)$ can be augmented with $3! = 6$ orders of span-tuples:

$$
\begin{cases}
R_{hr}(A, B, C) = (r_1, r_2), \\
R_{hr}(B, A, C) = (r_1^{-1}, r_2), \\
R_{hr}(A, C, B) = (r_2, r_1), \\
R_{hr}(B, C, A) = (r_2, r_1^{-1}), \\
R_{hr}(C, A, B) = (r_2^{-1}, r_1), \\
R_{hr}(C, B, A) = (r_1, r_2^{-1}).
\end{cases}
\tag{4}
$$

For other schemas, we can also obtain 6 fully-arranged cases of labeled span-tuples in a similar way, as described in Appendix A. If no n-ary relational fact exists between the three entities of span-tuples, then relation labels are set as null-label.

Since most cases of span-tuple are null-label, we set a weight hyperparameter $\alpha \in (0, 1]$ between the null-label and other labels to balance the learning of the null-label. We jointly trained the $n_r$ classifiers for each relations by cross-entropy loss $\mathcal{L}$ with a null-label weight bias $\mathbf{W}_\alpha$:

$$
\mathcal{L} = -\sum_{i=1}^{n_r} \mathbf{W}_\alpha \log \left( \frac{\exp\left(\mathbf{P}_i[r_i]\right)}{\sum_{j=1}^{|\mathcal{R}|+1} \exp\left(\mathbf{P}_{ij}\right)} \right),
\tag{5}
$$

where $\mathbf{W}_\alpha = [\alpha, 1.0, 1.0, ...1.0] \in \mathbb{R}^{(|\mathcal{R}|+1)}$.

| Dataset | #Ent | #R_hr | #R_ev | #R_ro | #R_hg | All | | Train | | Dev | | Test | |
|---------|------|-------|-------|-------|-------|-----|-----|-------|-----|-----|-----|------|-----|
| | | | | | | #Sentence | #Fact | #Sentence | #Fact | #Sentence | #Fact | #Sentence | #Fact |
| HyperRED | 40,293 | 106 | 232 | 168 | 62 | 44,840 | 45,994 | 39,840 | 39,978 | 1,000 | 1,220 | 4,000 | 4,796 |

Table 1: Dataset statistics, where the columns indicate the number of entities, relations with four schema, sentences and n-ary relational facts in all sets, train set, dev set, and test set, respectively.

## 4.4 Hetero-ordered Merging

In the decoding stage, since Text2NKG labels all 6 different arrangement of the same entity composition, we design a hetero-ordered merging strategy to merge the corresponding labels of these 6 hetero-ordered span-tuples into one to generate non-repetitive n-ary relational facts unsupervisedly. For *hyper-relational schema* ($n_r = 2$), we combine the predicted probabilities of two labels $\mathbf{P}_1, \mathbf{P}_2$ in 6 orders to $(A, B, C)$ order as follows:

$$\begin{cases} \mathbf{P}_1 = \mathbf{P}_1^{(ABC)} + I(\mathbf{P}_1^{(BAC)}) + \mathbf{P}_2^{(ACB)} \\ \quad\quad + I(\mathbf{P}_2^{(BCA)}) + \mathbf{P}_2^{(CAB)} + \mathbf{P}_1^{(CBA)}, \\ \mathbf{P}_2 = \mathbf{P}_2^{(ABC)} + \mathbf{P}_2^{(BAC)} + \mathbf{P}_1^{(ACB)} \\ \quad\quad + \mathbf{P}_1^{(BCA)} + I(\mathbf{P}_1^{(CAB)}) + I(\mathbf{P}_2^{(CBA)}), \end{cases} \quad (6)$$

where $I()$ is a function for swapping the predicted probability of relations and the corresponding inverse relations. Then, we take the maximum probability to obtain labels $r_1, r_2$, forming a 3-ary relational fact $(A, r_1, B, r_2, C)$ and filter it out if there are null-label in $(r_1, r_2)$. If there are inverse relation labels in $(r_1, r_2)$, we can also transform the order of entities and relations as equation 4. For *event-based schema*, *role-based schema*, and *hypergraph-based schema*, all can be generated by hetero-ordered merging according to this idea, as shown in Appendix B.

## 4.5 Output Merging

After hetero-ordered merging, we merge the output 3-ary relational facts to form higher-arity facts, with *hyper-relational schema* based on the same main triple, *event-based schema* based on the same main relation (event type), *role-based schema* based on the same key-value pairs, and *hypergraph-based schema* based on the same hyperedge relation. This way, we can unsupervisedly obtain n-ary relational facts with dynamic number of arity numbers for NKG construction. More details are discussed in Appendix G.2 and Appendix G.3.

## 5 Experiments

This section presents the experimental setup, results, and analysis. We answer the following research questions (RQs): **RQ1**: Does Text2NKG outperform other n-ary RE methods? **RQ2**: Whether Text2NKG can cover NKG construction for various schemas? **RQ3**: Does the main components of Text2NKG work? **RQ4**: How does the null-label bias hyperparameter in Text2NKG affect performance? **RQ5**: Can Text2NKG get complete n-ary relational facts in different arity? **RQ6**: How is Text2NKG's computational efficiency? **RQ7**: How does Text2NKG perform in specific case study? **RQ8**: What is the future development of Text2NKG in the era of large language models?

## 5.1 Experimental Setup

**Datasets.** The existing fine-grained n-ary RE dataset is **HyperRED** [5] only in *hyper-relational schema* with annotated extracted entities. Therefore, we expand the HyperRED dataset to four schemas as standard fine-grained n-ary RE benchmarks and conduct experiments on them. The statistics of the HyperRED with four schemas are shown in Table 1 and the construction detail is in Appendix C.

**Baselines.** We compare Text2NKG against **Generative Baseline** [14], **Pipeline Baseline** [24], and **CubeRE** [5] in fine-grained n-ary RE task of *hyper-relational schema*. For n-ary RE in the other three schemas, we compared Text2NKG with event extraction models such as **Text2Event** [16], **UIE** [17],

| Model | PLM | hyper-relational schema / Dev | | | hyper-relational schema / Test | | |
|---|---|---|---|---|---|---|---|
| | | Precision | Recall | $F_1$ | Precision | Recall | $F_1$ |
| **Unsupervised Method** | | | | | | | |
| ChatGPT | gpt-3.5-turbo | 12.0583 | 11.2764 | 11.6542 | 11.4021 | 10.9134 | 11.1524 |
| GPT-4 | gpt-4 | 15.7324 | 15.2377 | 15.4811 | 15.8187 | 15.4824 | 15.6487 |
| **Supervised Method** | | | | | | | |
| Generative Baseline | | 63.79 ± 0.27 | 59.94 ± 0.68 | 61.80 ± 0.37 | 64.60 ± 0.47 | 59.67 ± 0.35 | 62.03 ± 0.21 |
| Pipelinge Baseline | | 69.23 ± 0.30 | 58.21 ± 0.57 | 63.24 ± 0.44 | 69.00 ± 0.48 | 57.55 ± 0.19 | 62.75 ± 0.29 |
| CubeRE | | 66.14 ± 0.88 | 64.39 ± 1.23 | 65.23 ± 0.82 | 65.82 ± 0.84 | 64.28 ± 0.25 | 65.04 ± 0.29 |
| Text2NKG w/o DA | BERT-base (110M) | 76.02 ± 0.50 | 72.28 ± 0.68 | 74.10 ± 0.55 | 73.55 ± 0.81 | 70.63 ± 1.40 | 72.06 ± 0.34 |
| Text2NKG w/o $\alpha$ | | 88.77 ± 0.85 | 78.39 ± 0.47 | 83.26 ± 0.70 | 88.09 ± 0.69 | 76.64 ± 0.45 | 81.97 ± 0.58 |
| Text2NKG w/o HM | | 61.74 ± 0.34 | 76.97 ± 0.44 | 68.52 ± 0.69 | 61.07 ± 0.73 | 76.16 ± 0.59 | 67.72 ± 0.48 |
| Text2NKG (ours) | | **91.26 ± 0.69** | **79.36 ± 0.51** | **84.89 ± 0.44** | **90.77 ± 0.60** | **77.53 ± 0.32** | **83.63 ± 0.63** |
| Generative Baseline | | 67.08 ± 0.49 | 65.73 ± 0.78 | 66.40 ± 0.47 | 67.17 ± 0.40 | 64.56 ± 0.58 | 65.84 ± 0.25 |
| Pipelinge Baseline | BERT-large (340M) | 70.58 ± 0.78 | 66.58 ± 0.66 | 68.52 ± 0.32 | 69.21 ± 0.55 | 64.27 ± 0.24 | 66.65 ± 0.28 |
| CubeRE | | 68.75 ± 0.82 | 68.88 ± 1.03 | 68.81 ± 0.46 | 66.39 ± 0.96 | 67.12 ± 0.69 | 66.75 ± 0.28 |
| Text2NKG (ours) | | **91.90 ± 0.79** | **79.43 ± 0.42** | **85.21 ± 0.69** | **91.06 ± 0.81** | **77.64 ± 0.46** | **83.81 ± 0.54** |

Table 2: Comparison of Text2NKG with other baselines in the hyper-relational extraction on HyperRED. Results of the supervised baseline models are mainly taken from the original paper [5]. The best results in each metric are in **bold**.

| Model | PLM | event-based schema | | | role-based schema | | | hypergraph-based schema | | |
|---|---|---|---|---|---|---|---|---|---|---|
| | | Precision | Recall | $F_1$ | Precision | Recall | $F_1$ | Precision | Recall | $F_1$ |
| **Unsupervised Method** | | | | | | | | | | |
| ChatGPT | gpt-3.5-turbo | 10.4678 | 11.1628 | 10.8041 | 11.4387 | 10.4203 | 10.9058 | 11.2998 | 11.7852 | 11.5373 |
| GPT-4 | gpt-4 | 13.3681 | 14.6701 | 13.9888 | 13.6397 | 12.5355 | 13.0643 | 13.0907 | 13.6701 | 13.3741 |
| **Supervised Method** | | | | | | | | | | |
| Text2Event | | 73.94 ± 0.76 | 70.56 ± 0.58 | 72.21 ± 1.25 | 72.73 ± 0.79 | 68.45 ± 1.34 | 70.52 ± 0.62 | 73.68 ± 0.88 | 70.37 ± 0.51 | 71.98 ± 0.92 |
| UIE | T5-base (220M) | 76.51 ± 0.28 | 73.02 ± 0.66 | 74.72 ± 0.18 | 72.17 ± 0.29 | 69.84 ± 0.11 | 70.98 ± 0.31 | 72.03 ± 0.41 | 68.74 ± 0.13 | 70.34 ± 1.07 |
| LasUIE | | 79.62 ± 0.27 | 78.04 ± 0.75 | 78.82 ± 0.26 | 77.01 ± 0.20 | 74.26 ± 0.25 | 75.61 ± 0.24 | 76.21 ± 0.07 | 73.75 ± 0.17 | 74.96 ± 0.42 |
| Text2NKG | BERT-base (110M) | **86.20 ± 0.57** | **79.25 ± 0.33** | **82.58 ± 0.20** | **86.72 ± 0.80** | **78.94 ± 0.59** | **82.64 ± 0.38** | **83.53 ± 1.18** | **86.59 ± 0.38** | **85.03 ± 0.86** |
| Text2Event | | 75.58 ± 0.53 | 72.39 ± 0.82 | 73.97 ± 1.19 | 73.21 ± 0.45 | 70.85 ± 0.67 | 72.01 ± 0.31 | 75.28 ± 0.93 | 72.73 ± 1.07 | 73.98 ± 0.49 |
| UIE | T5-large (770M) | 79.38 ± 0.28 | 74.69 ± 0.61 | 76.96 ± 0.95 | 74.47 ± 1.42 | 71.84 ± 0.77 | 73.14 ± 0.38 | 74.57 ± 0.64 | 71.93 ± 0.86 | 73.22 ± 0.19 |
| LasUIE | | 81.29 ± 0.83 | 79.54 ± 0.26 | 80.40 ± 0.65 | 79.37 ± 0.92 | 76.63 ± 0.44 | 77.97 ± 0.76 | 77.49 ± 0.35 | 74.96 ± 0.60 | 76.20 ± 0.87 |
| Text2NKG | BERT-large (340M) | **88.47 ± 0.95** | **80.30 ± 0.75** | **84.19 ± 1.29** | **86.87 ± 0.87** | **80.86 ± 0.29** | **83.76 ± 1.17** | **85.06 ± 0.33** | **86.72 ± 0.36** | **85.89 ± 0.69** |

Table 3: Comparison of Text2NKG with other baselines in the n-ary RE in event-based, role-based, and *hypergraph-based schema*s on HyperRED. The best results in each metric are in **bold**.

and **LasUIE** [9]. Furthermore, we utilized different prompts to test the currently most advanced large-scale pre-trained language models **ChatGPT** [25] and **GPT-4** [19] in an unsupervised manner, specifically for the extraction performance across the four schemas. The detailed baseline settings can be found in Appendix D.

**Ablations.** To evaluate the significance of Text2NKG's three main components, data augmentation (DA), null-label weight hyperparameter ($\alpha$), and hetero-ordered merging (HM), we obtain three simplified model variants by removing any one component (**Text2NKG w/o DA**, **Text2NKG w/o** $\alpha$, and **Text2NKG w/o HM**) for comparison.

**Evaluation Metrics.** We use the $F_1$ score with precision and recall to evaluate the dev set and the test set. For a predicted n-ary relational fact to be considered correct, the entire fact must match the ground facts completely.

**Hyperparameters and Enviroment.** We train 10 epochs on HyperRED using the Adam optimizer. All experiments were done on a single NVIDIA A100 GPU, and all experimental results were derived by averaging 5 random seed experiments. Appendix E shows Text2NKG's optimal hyperparameter settings. Appendix F shows training details.

## 5.2 Main Results (RQ1)

The experimental results of proposed Text2NKG and other baselines with both BERT-base and BERT-large encoders can be found in Table 2 for the fine-grained n-ary RE in *hyper-relational schema*. We can observe that Text2NKG shows a significant improvement over the existing optimal model CubeRE on both the dev and test datasets of HyperRED. The $F_1$ score is improved by 19.66 percentage points in the dev set and 18.60 percentage points in the test set with the same BERT-base encoder, and 16.40 percentage points in the dev set and 17.06 percentage points in the test set with

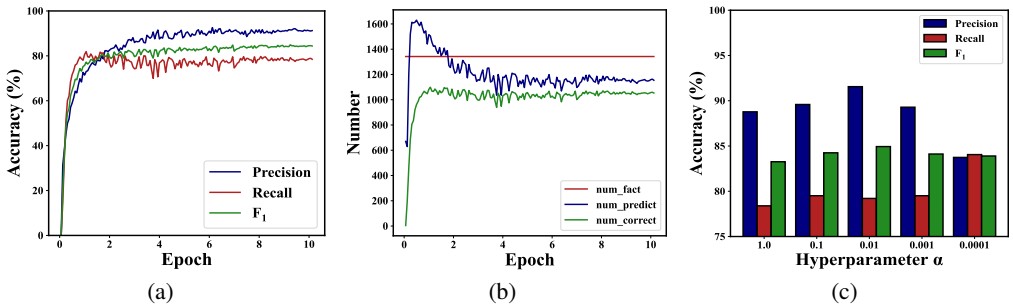

Figure 4: (a) Precision, Recall, and $F_1$ changes in the dev set during the training of Text2NKG. (b) The changes of the number of true facts, the number of predicted facts, and the number of predicted accurate facts during the training of Text2NKG. (c) Precision, Recall, and $F_1$ results on different null-label hyperparameter ($\alpha$) settings.

the same BERT-large encoder, reflecting Text2NKG's excellent performance. Figure 4(a) and 4(b) intuitively show the changes of evaluation metrics and answers of facts in the dev set during the training of Text2NKG. It is worth noting that Text2NKG exceeds 90% in precision accuracy, which proves that the model can obtain very accurate n-ary relational facts and provides a good guarantee for the quality of fine-grained NKG construction.

## 5.3 Results on Various NKG Schemas (RQ2)

As shown in Table 3, besides *hyper-relational schema*, Text2NKG also accomplishes the tasks of fine-grained n-ary RE in three other different NKG schemas on HyperRED, which demonstrates good utility. In the added tasks of n-ary RE for event-based, role-based, and *hypergraph-based schema*s, since no model has done similar experiments at present, we used event extraction or unified extraction methods such as Text2Event [16], UIE [17], and LasUIE [9] for comparison. Text2NKG still works best in these schemas, which demonstrates good versatility.

## 5.4 Ablation Study (RQ3)

Data augmentation (DA), null-label weight hyperparameter ($\alpha$), and hetero-ordered merging (HM) are the three main components of Text2NKG. For the different Text2NKG variants as shown in Table 2, DA, $\alpha$, and HM all contribute to the accurate results of our complete model. By comparing the differences, we find that HM is most effective by combining the probabilities of labels of different orders, followed by DA and $\alpha$.

## 5.5 Analysis of Null-label Weight Hyperparameters (RQ4)

We compared the effect for different null-label weight hyperparameters ($\alpha$). As shown in Figure 4(c), the larger the $\alpha$, the greater the learning weight of null-label compared with other lables, the more relations are predicted as null-label. After filtering out the facts having null-label, fewer facts are extracted, so the precision is generally higher, and the recall is generally lower. The smaller the $\alpha$, the more relations are predicted as non-null labels, thus extracting more n-ary relation facts, so the recall is generally higher, and the precision is generally lower. Comparing the results of $F_1$ values for different $\alpha$, it is found that $\alpha = 0.01$ works best, which can be adjusted in practice according to specific needs to obtain the best results.

## 5.6 Analysis of N-ary Relation Extraction in Different Arity (RQ5)

Figure 5(a) shows the number of n-ary relational facts extracted after output merging and the number of the answer facts in different arity during training of Text2NKG on the dev set. We find that, as the training proceeds, the final output of Text2NKG converges to the correct answer in terms of the number of complete n-ary relational facts in each arity, achieving implementation of n-ary RE in indefinite arity unsupervised, with good scalability.

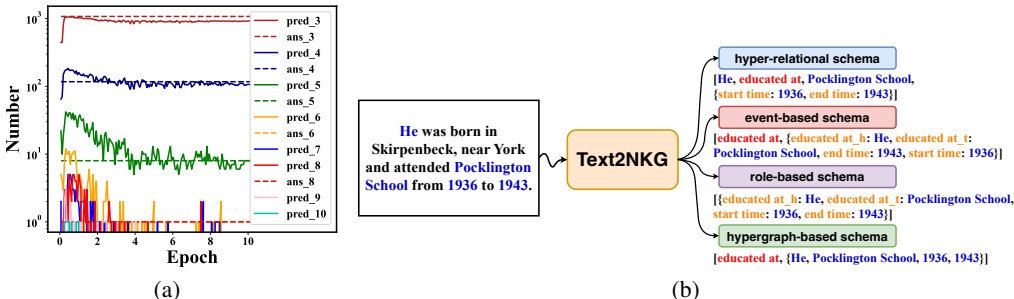

Figure 5: (a) The changes of the number of extracted n-ary RE in different arity, where "pred_n" represents the number of extracted n-ary facts with different arities by Text2NKG, and "ans_n" represents the ground truth. (b) Case study of Text2NKG's n-ary relation extraction in four schemas on HyperRED.

## 5.7 Computational Efficiency (RQ6)

As mentioned in Section 4.2, the main computational consumption of Text2NKG is selecting every span-tuple of three ordered entities to encode them and get the classified labels in multiple-label classification part. If we adopt an traversal approach with each span-tuple in one training items, the time complexity will be O($m^3$). To reduce the high time complexity of training every span-tuple as one training item, Text2NKG uses packed levitated markers that pack one training item with each entity in $\mathcal{E}_s$ separately. We obtain the embedding representation of the three entities to form $A_{m-1}^2$ span-tuples in one training item. Every input sentence contains $m$ training items with $mA_{m-1}^2 = A_m^3$ span-tuples for any ordered arrangement of three entities for multiple-label classification. Therefore, the time complexity decreased from O($m^3$) to O($m$).

## 5.8 Case Study (RQ7)

Figure 5(b) shows a case study of n-ary RE by a trained Text2NKG. For a sentence, `"He was born in Skirpenbeck, near York and attended Pocklin."`, four structured n-ary RE can be obtained by Text2NKG according to the requirements. Taking the *hyper-relational schema* for an example, Text2NKG can successfully extract one n-ary relational fact consisting of a main triple `[He, educated at, Pocklington]`, and two auxiliary key-value pairs `{start time:1936}`,`{end time:1943}`. This intuitively validates the practical performance of Text2NKG on fine-grained n-ary RE to better contribute to NKG construction.

## 5.9 Comparison with ChatGPT (RQ8)

As shown in Table 2 and Table 3, we compared the extraction effects under four NKG schemas of the supervised Text2NKG with the unsupervised ChatGPT and GPT-4. We found that these large language models cannot accurately distinguish the closely related relations in the fine-grained NKG relation repository, resulting in their F1 scores ranging around 10%-15%, which is much lower than the performance of Text2NKG. On the other hand, the limitation of Text2NKG is that its performance is confined within the realm of supervised training. Therefore, in future improvements and practical applications, we suggest combining small supervised models with large unsupervised models to balance solving the cold-start and fine-grained extraction, which is detailed in Appendix G.1.

## 6 Conclusion

In this paper, we introduce Text2NKG, a novel framework designed for fine-grained n-ary relation extraction (RE) aimed at constructing N-ary Knowledge Graphs (NKGs). Our extensive experiments demonstrate that Text2NKG outperforms all existing baseline models across a wide range of fine-grained n-ary RE tasks. Notably, it excels in four distinct schema types: hyper-relational, event-based, role-based, and hypergraph-based. Furthermore, we have extended the HyperRED dataset, transforming it into a comprehensive fine-grained n-ary RE benchmark that supports all four schemas.

## Acknowledgments

This work is supported by the National Science Foundation of China (Grant No. 62176026, Grant No. 62406036, Grant No. 62473271, and Grant No. 62076035). This work is also supported by the SMP-Zhipu.AI Large Model Cross-Disciplinary Fund, the BUPT Excellent Ph.D. Students Foundation (No. CX2023133), the BUPT Innovation and Entrepreneurship Support Program (No. 2024-YC-A091 and No. 2024-YC-T022), and the Engineering Research Center of Information Networks, Ministry of Education.

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

# Appendix

## A  Supplement to Data Augmentation

In addition to the *hyper-relational schema*, the data augmentation strategies for other schemas are as follows:

For *event-based schema*, given an event-based fact $(r_1, r_2, A, r_3, B, r_4, C)$, we consider keeping the main relation $r_1$ unchanged, and swapping other key-value pairs, $\{r_2, A\}$, $\{r_3, B\}$, and $\{r_4, C\}$, positionally, also as labeled training span-tuple cases. Thus $R_{ev}(A, B, C) = (r_1, r_2, r_3, r_4)$ can be augmented with 6 orders of span-tuples:

$$
\begin{cases}
R_{ev}(A, B, C) = (r_1, r_2, r_3, r_4), \\
R_{ev}(B, A, C) = (r_1, r_3, r_2, r_4), \\
R_{ev}(A, C, B) = (r_1, r_2, r_4, r_3), \\
R_{ev}(B, C, A) = (r_1, r_3, r_4, r_2), \\
R_{ev}(C, A, B) = (r_1, r_4, r_2, r_3), \\
R_{ev}(C, B, A) = (r_1, r_4, r_3, r_2).
\end{cases}
\tag{7}
$$

For *role-based schema*, given a role-based fact $(r_1, A, r_2, B, r_3, C)$, we consider swapping key-value pairs, $\{r_1, A\}$, $\{r_2, B\}$, and $\{r_3, C\}$, positionally, also as labeled training span-tuple cases. Thus $R_{ro}(A, B, C) = (r_1, r_2, r_3)$ can be augmented with 6 orders of span-tuples:

$$
\begin{cases}
R_{ro}(A, B, C) = (r_1, r_2, r_3), \\
R_{ro}(B, A, C) = (r_2, r_1, r_3), \\
R_{ro}(A, C, B) = (r_1, r_3, r_2), \\
R_{ro}(B, C, A) = (r_2, r_3, r_1), \\
R_{ro}(C, A, B) = (r_3, r_1, r_2), \\
R_{ro}(C, B, A) = (r_3, r_2, r_1).
\end{cases}
\tag{8}
$$

For *hypergraph-based schema*, given a hypergraph-based fact $(r_1, A, B, C)$, we consider keeping the main relation $r_1$ unchanged, and swapping entities, $A$, $B$, and $C$, positionally, also as labeled training span-tuple cases. Thus $R_{hg}(A, B, C) = (r_1)$ can be augmented with 6 orders of span-tuples:

$$
\begin{cases}
R_{hg}(A, B, C) = (r_1), \\
R_{hg}(B, A, C) = (r_1), \\
R_{hg}(A, C, B) = (r_1), \\
R_{hg}(B, C, A) = (r_1), \\
R_{hg}(C, A, B) = (r_1), \\
R_{hg}(C, B, A) = (r_1).
\end{cases}
\tag{9}
$$

## B  Supplement to Hetero-ordered Merging

In addition to the *hyper-relational schema*, the hetero-ordered merging strategies for other schemas are as follows:

For *event-based schema* ($n_r = 4$), we combine the predicted probabilities of four labels $\mathbf{P}_1, \mathbf{P}_2, \mathbf{P}_3, \mathbf{P}_4$ in 6 orders to $(A, B, C)$ order as follows:

$$
\begin{cases}
\mathbf{P}_1 = \mathbf{P}_1^{(ABC)} + \mathbf{P}_1^{(BAC)} + \mathbf{P}_1^{(ACB)} \\
\qquad + \mathbf{P}_1^{(BCA)} + \mathbf{P}_1^{(CAB)} + \mathbf{P}_1^{(CBA)}, \\
\mathbf{P}_2 = \mathbf{P}_2^{(ABC)} + \mathbf{P}_3^{(BAC)} + \mathbf{P}_2^{(ACB)} \\
\qquad + \mathbf{P}_4^{(BCA)} + \mathbf{P}_3^{(CAB)} + \mathbf{P}_4^{(CBA)}, \\
\mathbf{P}_3 = \mathbf{P}_3^{(ABC)} + \mathbf{P}_2^{(BAC)} + \mathbf{P}_4^{(ACB)} \\
\qquad + \mathbf{P}_2^{(BCA)} + \mathbf{P}_4^{(CAB)} + \mathbf{P}_3^{(CBA)}, \\
\mathbf{P}_4 = \mathbf{P}_4^{(ABC)} + \mathbf{P}_4^{(BAC)} + \mathbf{P}_3^{(ACB)} \\
\qquad + \mathbf{P}_3^{(BCA)} + \mathbf{P}_2^{(CAB)} + \mathbf{P}_2^{(CBA)}.
\end{cases}
\tag{10}
$$

Then, we take the maximum probability to obtain labels $r_1, r_2, r_3, r_4$, forming a 3-ary relational fact $(r_1, r_2, A, r_3, B, r_4, C)$ and filter it out if there are null-label in $(r_1, r_2, r_3, r_4)$.

For *role-based schema* ($n_r = 3$), we combine the predicted probabilities of three labels $\mathbf{P}_1, \mathbf{P}_2, \mathbf{P}_3$ in 6 orders to $(A, B, C)$ order as follows:

$$
\begin{cases}
\mathbf{P}_1 = \mathbf{P}_1^{(ABC)} + \mathbf{P}_2^{(BAC)} + \mathbf{P}_1^{(ACB)} \\
\qquad + \mathbf{P}_3^{(BCA)} + \mathbf{P}_2^{(CAB)} + \mathbf{P}_3^{(CBA)}, \\
\mathbf{P}_2 = \mathbf{P}_2^{(ABC)} + \mathbf{P}_1^{(BAC)} + \mathbf{P}_3^{(ACB)} \\
\qquad + \mathbf{P}_1^{(BCA)} + \mathbf{P}_3^{(CAB)} + \mathbf{P}_2^{(CBA)}, \\
\mathbf{P}_3 = \mathbf{P}_3^{(ABC)} + \mathbf{P}_3^{(BAC)} + \mathbf{P}_2^{(ACB)} \\
\qquad + \mathbf{P}_2^{(BCA)} + \mathbf{P}_1^{(CAB)} + \mathbf{P}_1^{(CBA)}.
\end{cases}
\tag{11}
$$

Then, we take the maximum probability to obtain labels $r_1, r_2, r_3$, forming a 3-ary relational fact $(r_1, A, r_2, B, r_3, C)$ and filter it out if there are null-label in $(r_1, r_2, r_3)$.

For *hypergraph-based schema* ($n_r = 1$), we combine the predicted probabilities of one label $\mathbf{P}_1$ in 6 orders to $(A, B, C)$ order as follows:

$$
\begin{cases}
\mathbf{P}_1 = \mathbf{P}_1^{(ABC)} + \mathbf{P}_1^{(BAC)} + \mathbf{P}_1^{(ACB)} \\
\qquad + \mathbf{P}_1^{(BCA)} + \mathbf{P}_1^{(CAB)} + \mathbf{P}_1^{(CBA)}.
\end{cases}
\tag{12}
$$

Then, we take the maximum probability to obtain labels $r_1$, forming a 3-ary relational fact $(r_1, A, B, C)$ and filter it out if $r_1$ is null-label.

## C  Construction of Dataset

Based on the original *hyper-relational schema* on HyperRED dataset [5], we construct other three schemas (event-based, role-based, and hypergraph-based) for fine-grained n-ary RE. Firstly, we view the main relation in the *hyper-relational schema* as the event type in the *event-based schema*, combine the head entity and tail entity with two extra head key and tail key to convert them into two key-value pairs, and remain the auxiliary key-value pairs in the *hyper-relational schema*. Taking '*Einstein received his Doctorate degree in Physics from the University of Zurich.*' as an example, it can be represented as (*Einstein, educated, University of Zurich, {academic_major, Physics}, {academic_ degree, Doctorate}*) in the *hyper-relational schema* and (*education, {trigger, received}, {person, Einstein}, {college, University of Zurich}, {academic_major, Physics},{academic_degree, Doctorate}*) in the *event-based schema*. Secondly, we remove the event type in the *event-based schema* to obtain the *role-based schema*. Thirdly, we remove all the keys in key-value pairs and remain the relation to build the *hypergraph-based schema*.

## D    Baseline Settings

Firstly, for the original *hyper-relational schema* of HyperRED, we adopted the same baselines as in the CubeRE paper [5] to compare with Text2NKG:

**Generative Baseline:**    Generative Baseline uses BART [14], a sequence-to-sequence model, to transform input sentences into a structured text sequence.

**Pipeline Baseline:**    Pipeline Baseline uses UniRE [24] to extract relation triplets in the first stage and a span extraction model based on BERT-Tagger [6] to extract value entities and corresponding qualifier labels in the second stage.

**CubeRE:**    CubeRE [5] is the only hyper-relational extraction model that uses a cube-filling model inspired by table-filling approaches and explicitly considers the interaction between relation triplets and qualifiers.

Secondly, for the *event-based schema*, *role-based schema*, and *hypergraph-based schema*, we added the following baselines to further validate the effect of Text2NKG on the fine-grained N-ary relation fact extraction task in the HyperRED dataset:

**Text2Event:**    Text2Event [5] is a classic model in the Event extraction domain. However, it is not applicable to extractions of the *hyper-relational schema*. For the *role-based schema* extraction, we retained the key without referring to the main relation, while for the *hypergraph-based schema* extraction, we retained the main relation without referring to the key to get the final result for comparison.

**UIE / LasUIE:**    UIE [17] and LasUIE [9] are unified information extraction models that can handle most tasks like NER, RE, EE, etc.  However, they are still only suitable for event extraction in the multi-relational extraction domain and are not applicable to extractions of the *hyper-relational schema*. Therefore, we adopted the same approach as with Text2Event to compare with Text2NKG.

Thirdly, under the impact of the wave of large-scale language models brought about by ChatGPT on traditional natural language processing tasks, we added unsupervised large models as baselines to compare with Text2NKG in the n-ary RE tasks of the four schemas.

**ChatGPT / GPT4:**    Using different prompts, we tested the latest state-of-the-art large-scale pre-trained language models ChatGPT [25] and GPT-4 [19] in an unsupervised manner, evaluating their performance on the extraction of the four schemas.

## E    Hyperparameter Settings

We use the grid search method to select the optimal hyperparameter settings for both Text2NKG with Bert-base and Bert-large. We use the same hyperparameter settings in Text2NKG with different encoders. The hyperparameters that we can adjust and the possible values of the hyperparameters are first determined according to the structure of our model in Table 4.  Afterward, the optimal hyperparameters are shown in **bold**.

| Hyperparameter | HyperRED |
|:---:|:---:|
| $\alpha$ | $\{1.0, 0.1, \mathbf{0.01}, 0.001\}$ |
| Train_batch_size | $\{2, 4, \mathbf{8}, 16\}$ |
| Eval_batch_size | $\{\mathbf{1}\}$ |
| Learning rate | $\{1e-5, \mathbf{2e\text{-}5}, 5e-5\}$ |
| Max_sequence_length | $\{128, \mathbf{256}, 512, 1024\}$ |
| Weight decay | $\{\mathbf{0.0}, 0.1, 0.2, 0.3\}$ |

Table 4:  Hyperparameter Selection.

## F    Model Training Details

We train 10 epochs on HyperRED with the optimal combination of hyperparameters. Text2NKG and all its variants have been trained on a single NVIDIA A100 GPU. Using our optimal hyperparameter settings, the time required to complete the training on HyperRED is 4h with BERT-base encoder and 10h with BERT-large encoder.

# G Further Discussions

## G.1 How does ChatGPT perform in Fine-grained N-ary RE tasks?

We have tried to use LLM APIs such as ChatGPT and GPT to do similar n-ary RE tasks, i.e., prompting model input and output formats for extraction. The advantage of ChatGPT is that it can perform similar tasks in a few-shot situation, however, for building high-quality knowledge graphs, the performance and the fineness of the n-ary RE are much lower than Text2NKG. This is because ChatGPT is not good at multi-label classification tasks that contain less semantic interpretation. When the number of labels of relations in our relation collection is very large, we need to write a very long prompt to tell the LLM about our label candidate collection, which again leads to the problem of forgetting. Therefore, we have tried numerous prompt templates to enhance the extraction effect of ChatGPT, however, on fine-grained n-ary RE task, the best result of ChatGPT can only reach about 10% of $F_1$ value on HyperRED, which is much lower than the result of 80%+ $F_1$ value of Text2NKG.

However, advanced LLMs such as ChatGPT are a good idea for training dataset generation for Text2NKG in such tasks to save some manual labor to only verify and correct the training items generated. For future work, we will continue our research in this direction and try to combine large language models with Text2NKG-like supervised models for automated fine-grained n-ary RE for n-ary relational knowledge graph construction.

## G.2 Why first Extracting 3-ary facts and then Merging them into N-ary Facts ?

We use output merging to address the dynamic changes in the number of elements in n-ary relational facts. The atomic unit of an n-ary fact includes a 3-ary fact with three entities. For instance, in the hyper-relational fact (*Einstein, educated_at, University of Zurich, degree: Doctorate degree, major: Physics*), the Text2NKG algorithm allows us to extract two 3-ary atomic facts: (*Einstein, educated_at, University of Zurich, degree: Doctorate degree*) and (*Einstein, educated_at, University of Zurich, major: Physics*). These are then merged based on the same primary triple (*Einstein, educated_at, University of Zurich*) to form a 4-ary fact. The same principle applies to facts of higher arities.

As another example demonstrating the problem with merging binary relations: consider the statement "*Einstein received his Bachelor's degree in Mathematics and his Doctorate degree in Physics.*" When represented as binary relations, the facts become (*Einstein, degree, Doctorate degree*), (*Einstein, major, Physics*), (*Einstein, degree, Bachelor*), and (*Einstein, major, Mathematics*). With this representation, we cannot merge these binary relation facts effectively because there's no way to determine whether *Einstein*'s *doctoral major* was *Physics* or *Mathematics*. This necessitates the use of NKG's n-ary relationship facts to represent this information, as seen in (*Einstein, degree, Doctorate degree, major, Physics*).

Therefore, using binary facts, we can't merge them into n-ary facts based on shared elements within these facts. On the other hand, using facts with four entities or more makes it challenging to effectively extract 3-ary atomic facts.

In Section 5.6 and Figure 5(a), we also analyzed the effects and detailed insights of unsupervised extraction of arbitrary-arity facts.

## G.3 How Text2NKG can address Long Contexts with Relations spread across Various Sentences ?

As long as the text to be extracted is a lengthy piece with entities annotated, it can undergo long-form n-ary relation extraction. The maximum text segment size for our proposed method depends on the maximum text length that a transformer-based encoder can accept, such as Bert-base and Bert-large, which have a maximum limit of 512. To extract from larger documents, we simply need to switch to encoders with larger context length, which all serve as the encoder portion of Text2NKG and are entirely decoupled from the n-ary relation extraction technique we propose. This is one of the advantages of Text2NKG. Its primary focus is to address the order and combination issues of multi-ary relationships. We can seamlessly combine a transformer encoder that supports long texts with Span-tuple Multi-label Classification to process n-ary relation extraction in long chapters.

