# OpenReview forum: "Text2NKG: Fine-Grained N-ary Relation Extraction for N-ary relational Knowledge Graph Construction"
_NeurIPS.cc/2024/Conference — NeurIPS 2024 poster_

### Official Review · Reviewer_XsBb · 2024-07-09

**Soundness:** 2
**Presentation:** 3
**Contribution:** 3
**Rating:** 6
**Confidence:** 4

**Summary:**

This work introduces Text2NKG. A proclaimed first method of extracting n-ary facts from text for construction a KG. The focus on n-ary is a more complex task than standard RE as an n-ary fact can hold more entities than the standard RDF  {subject, relation, object}.

Candidate 3-ary span tuples are formed from entities combinations in a sentence. This is embedded using BERT, and put through linear classifiers (Feed-Forward-Networks). The resulting 3-ary facts are then combined for n-ary facts using "hetero-ordered merging". The fine-tuning process makes use of a data augmentation technique, a balancing parameter to compensate for the large amount of "no relation" labels, and without merging.

The method is verified on HyperRED (the only public hyperrelational dataset currently available), and beats several SOTA methods.

**Strengths:**

- The motivation and contributions of the paper are clear and concrete. The previous limitations of research (Diversity, determination of order, and variability) are clearly mentioned and addressed.
	- The authors compare to a variety of different baselines, including SOTA LLM (GPT).
	- For this, a number of prompt templates have been considered
	- Despite the sometimes overwhelming amount of notations, the work is generally pleasant to read.

**Weaknesses:**

- The provided code is complex with low readibility (>5 nested for/while statements with little/no comments and no documentation).
	- Code makes mention of ACE dataset, but this does not seem to have made it into the end-work, adding to the confusion.
	- Other than the case study in Figure 5, the error analysis is fairly weak and generally performance based. There is no inspection provided as to what explains (for example) the discrepancy between "Text2NKG" and "Text2NKG w/o HM".
	- Prompt templates for GPT that were considered (or eventually used for best result) are not included
	- BERT was trained on a copy of Wikipedia in 2018. There seems to be no consideration that the data in HyperRED may overlap with the pre-training data of BERT.
	- The work hyperfixates on a single benchmark: HyperRED, despite others being mentioned.



Comments:
Figure 2: Typo "hyper-relaional"
Line 159: typo
Line 214: Broken sentence.
Line 219: Why was it decided to name these Generative and Pipeline and not specific to the previous works? (BART and UniRE)
Table 2: The GPT results dont need 4 decimal places
Table 2: typo (pipelinge)

Equation 6: It is unclear to me how this is derived, and it is not explained why (the how is there) this specific formulation works.
Figure 5a: Figure is hard to read.

**Questions:**

Questions:
	- Line 40: Several more benchmarks for NKG are mentioned (JF17K, Wikipeople, WD50K, etc.), but not considered (or no argumentation is provided). Why were these not considered?
	- From inspection, what are cases where the model fails (e.g. seemingly difficult cases where it struggles) or cases where it consistenly succeeds?

**Limitations:**

Yes

---

> ### Author Rebuttal · Authors · 2024-08-02
>
> Thank you for a careful review of our work. We appreciate that you find our work concrete in motivation and pleasant to read. We respond to your specific concerns and questions below.
>
> ---
> **For Weakness 1 (Code makes mention of ACE dataset, but this does not seem to have made it into the end-work.):**
>
> We are pleased for your interest in our code. We will optimize it for better contribution to the open-source community. Previously, we converted the ACE dataset as a benchmark for n-ary relation extraction, but reviewers found it contributed little as a benchmark, so we removed it from the paper. Any remaining mentions in the code will be corrected in the final release. Thank you for your feedback.
>
> ---
> **For Weakness 2 (There is no inspection provided as to what explains the discrepancy between "Text2NKG" and "Text2NKG w/o HM".):**
>
> Thanks for your suggestions. As described in **Section 3**, n-ary RE involves more entity orders than binary RE. Text2NKG uses a hetero-ordered merging method to consider probabilities of different entity orders.
>
> In the **Text2NKG w/o HM** setting, we didn't combine probabilities of different arrangements during decoding, leading to a decrease in performance.
>
> ---
> **For Weakness 3 (Prompt templates for GPT that were considered are not included):**
>
> Thanks for your suggestions. Our ChatGPT and GPT-4 prompts both adopted a 1-shot unsupervised design. The specific input prompts (which we will add to the Appendix of the final paper) can be referenced from our response to the **Weakness 1(i)** mentioned by **Reviewer rUhA**.
>
> ---
> **For Weakness 4 (There seems to be no consideration that the data in HyperRED may overlap with the pre-training data of BERT.):**
>
> BERT is widely used for downstream tasks, including n-ary relation extraction. Baseline models like CubeRE and those from the HyperRED dataset paper are also fine-tuned on BERT-based pre-trained methods for fair comparison. Plus, since Wikipedia is not originally structured with n-ary relational schemas, HyperRED focuses on extracting these structured relations from unstructured text, making it a more challenging task that involves entity count, order, and evaluation criteria.
>
> ---
> **For Weakness 5 (The work hyperfixates on a single benchmark: HyperRED, despite others being mentioned.):**
>
> There is a scarcity of work on automatically constructing NKGs from natural texts. HyperRED is currently the only hyper-relational extraction dataset. Text2NKG is the first method to unify n-ary RE extraction for four types of NKG schemas and can support more NKG schemas.
>
> ---
> **For Comment (1) (Typos in Figure 2, Line 159, Line 214, and Table 2.):**
>
> Thanks for your carefully reading. We will correct "hyper-relaional" in **Figure 2** to "hyper-relational", "imput" in **Line 159** to "input", 4 decimal places in **Table 2** to 2 decimal places, "Pipelinge" in **Table 2** to "Pipeline". Your proposal greatly enhances the presentation of the paper and will be all implemented.
>
> ---
> **For Comment (2) (Why was it decided to name these Generative and Pipeline?):**
>
> As stated in **Table 2**, the results of the supervised baseline models are primarily taken from the original paper of CubeRE. For consistency, we retained the baseline names from the CubeRE paper, using the same terms: Generative and Pipeline instead of BART and UniRE.
>
> ---
> **For Comment (3) (Equation 6: It is unclear to me how this is derived, and it is not explained why this specific formulation works.):**
>
> In the data augmentation section in **Section 4.3**, we trained on all 6 arrangements of A, B, C. Therefore, in Hetro-ordered Merging (prediction phase), we also need to consider all 6 arrangements to accurately evaluate (A, B, C). However, we must convert the 6 different orders back to $(A, r_1, B, r_2, C)$ to obtain the corresponding probability $P_i$ for each $r_i$ and sum them up. Let's take the calculation of $\mathbf{P}_{1}$ as an example, which is derived from the sum of six terms. The first term $\mathbf{P}_1^{(ABC)}$ represents the probability of $r_1$ in $(A, r_1, B, r_2, C)$ without any operation. The second term $I(\mathbf{P}_1^{(BAC)})$ represents the probability of $r_1'=r_1^{-1}$ in $(B, r_1^{-1}, A, r_2, C)$, where $r_1'$ needs to be inverted to transform to (A, B, C). The remaining 4 terms in the equation follow the similar logic. Finally, we calculate the combined probability of the two $r_1,r_2$ in (A, B, C) to get Equation 6.
>
> ---
> **For Comment (4) (Figure 5a is hard to read.):**
>
> In **Figure 5a**, "pred_n" represents the number of extracted n-ary facts with different arities $n$ by Text2NKG, and "ans_n" represents the ground truth. As training progresses, Text2NKG's output converges to the correct number of n-ary facts for each arity, demonstrating its ability to handle n-ary RE with arbitrary arity and good scalability.
>
> ---
> **For Question 1 (Several more benchmarks for NKG are mentioned (JF17K, Wikipeople, WD50K, etc.). Why were these not considered?):**
>
> Datasets like JF17K, Wikipeople, and WD50K, used for link prediction in n-ary relational knowledge graphs (NKGs), lack original text. Practical fields like medicine, finance, and law require extracting n-ary relational facts from natural language. HyperRED is the only high-quality dataset for this. Mentioning other NKGs helps design Text2NKG's method to cover all NKG schemas.
>
> ---
> **For Question 2 (What are cases where the model fails or cases where it consistenly succeeds?):**
>
> From **Table 2** and **Table 3**, Text2NKG shows higher Precision than Recall, indicating it accurately extracts n-ary facts but often misses some.
>
> To balance this, we introduced a null-label weight hyperparameter ($\alpha$). As shown in Figure 4(c), increasing $\alpha$ raises Precision but lowers Recall by predicting more null-labels. Decreasing $\alpha$ does the opposite. This adjustability allows Text2NKG to be optimized for different scenarios.

---

> > ### Comment · Reviewer_XsBb · 2024-08-12
> >
> > The answer for question 2 still makes for a quantitative only-evaluation with little to no qualitative inspection (by looking at examples where the model does well or doesnt do well).
> >
> > I decide to keep my previous scores.

---

### Official Review · Reviewer_GMbk · 2024-07-10

**Soundness:** 4
**Presentation:** 4
**Contribution:** 4
**Rating:** 7
**Confidence:** 5

**Summary:**

Text2NKG is a novel framework for fine-grained n-ary relation extraction aimed at constructing n-ary relational knowledge graphs (NKGs). Unlike traditional binary relational knowledge graphs, NKGs encompass relations involving more than two entities, making them more reflective of real-world scenarios. Text2NKG employs a span-tuple classification approach along with hetero-ordered merging and output merging techniques. It supports multiple NKG schemas—hyper-relational, event-based, role-based, and hypergraph-based—and achieves state-of-the-art performance in fine-grained n-ary relation extraction benchmarks.

**Strengths:**

1) Traditional triplet representations of knowledge cannot fully express complex relationships. However, there is limited research on the extraction of multi-ary and hyper-relations. This study on fine-grained n-ary relations has significant potential applications.
2) The method proposed in the article improved extraction effectiveness by 20% in n-ary extraction.
3) The author considered many additional technical issues, such as comparisons of LLMs and handling long texts, and conducted supplementary analyses.
4) A clever method was used to solve the entity order problem in n-ary relation extracion, and the experiments demonstrated that it is possible to extract facts with varying numbers of arity in an unsupervised manner.
5) The code and datasets used in the experiments are made publicly available, promoting transparency, reproducibility, and further research in this area.

**Weaknesses:**

1) The comparison settings with the large language models need to be clearer.
2) Line 159: imput -> input.

**Questions:**

What is the connection and difference between n-ary relation extraction and event extraction?

**Limitations:**

The authors have listed the limitations in the Checklist.

---

> ### Author Rebuttal · Authors · 2024-08-02
>
> Thank you for a detailed review. We are delighted that you think our work have significant potential applications and technical effectiveness. We answer your questions below.
>
> ---
> **For Weakness 1 (The comparison settings with the large language models need to be clearer.):**
>
> Thanks for your suggestions. Our ChatGPT and GPT-4 prompts both adopted a 1-shot unsupervised design. The specific input prompts (which we will add to the Appendix of the final paper) can be referenced from my response to the **Weakness 1(i)** mentioned by **Reviewer rUhA**.
>
> ---
> **For Weakness 2 (Line 159: imput -> input.):**
>
> Thanks for your carefully reading. We will correct this typo in the final paper.
>
> ---
> **For Question (What is the connection and difference between n-ary relation extraction and event extraction?):**
>
> N-ary relation extraction and event extraction are closely related tasks in natural language processing that aim to extract structured information from text. N-ary relation extraction focuses on identifying relationships involving more than two entities. Event extraction, as one of specific schema in n-ary relation extraction, identifies events described in the text and extracts participants and attributes associated with these events. An event consists of a trigger word indicating the occurrence and arguments that represent the entities involved.
>
> The primary difference between the two lies in their focus and output. N-ary relation extraction captures static relationships among multiple entities, resulting in tuples that represent complex interactions. Event extraction, however, focuses on dynamic occurrences, identifying events, triggers, and arguments to produce structured event representations.

---

> > ### Comment · Reviewer_GMbk · 2024-08-12
> >
> > Thank you for your clear explanations. Your responses have addressed my concerns, and I appreciate the additional insights provided. I maintain my positive decision.

---

### Official Review · Reviewer_ED7F · 2024-07-10

**Soundness:** 4
**Presentation:** 3
**Contribution:** 4
**Rating:** 7
**Confidence:** 4

**Summary:**

The paper introduces a novel framework for fine-grained n-ary relation extraction aimed at constructing n-ary relational knowledge graphs (NKGs). Traditional KGs primarily focus on binary relations, but this work targets n-ary relations which involve more than two entities, aligning more closely with real-world facts.

**Strengths:**

1. The writing of the paper is clear, well-structured. The logical flow from the problem definition to the proposed methodology and experimental results makes it easy to follow.
2. The span-tuple classification and hetero-ordered merging approach are novel and effective, enabling the extraction of fine-grained n-ary relations which are more representative of real-world facts compared to traditional binary relations.
3. Proposed method achieves state-of-the-art performance on the HyperRED benchmark, with significant improvements in F1 scores over existing methods.
4. The paper compares Text2NKG with ChatGPT (gpt-3.5-turbo) and GPT-4 for N-ary extraction performance across the four schemas, and analyzes the advantages and disadvantages between large and small models for NKG construction.

**Weaknesses:**

1. Although the framework shows potential for scalability, the paper does not thoroughly address the computational efficiency and scalability when applied to real-time applications.
2. While the paper includes ablation studies, more detailed analysis and discussion on the impact of each component (e.g., data augmentation, null-label weight hyperparameter) on different types of NKG schemas would provide deeper insights into their contributions.

**Questions:**

1. Can you provide an example of extracting N-ary relations in real-world information extraction?
2. How should NKG be stored and utilized, and can n-ary extraction be applied in RAG or Agent in the era of LLMs?

**Limitations:**

See the Weakness.

---

> ### Author Rebuttal · Authors · 2024-08-02
>
> Thank you for your insightful evaluation of our paper. We are happy that you find our paper clear to follow and the results impressive. We respond to your concerns and questions below.
>
> ---
> **For Weakness 1 (The paper does not thoroughly address the computational efficiency and scalability.):**
>
> As described in **Section 3** and **Section 4.2**, given an input sentence with $l$ words $s=\{w_1,w_2,...,w_l\}$, an entity $e$ is a consecutive span of words: $e=\{w_p,w_{p+1},...,w_q\}\in \mathcal{E}_s$, where $p,q\in\{1,...,l\}$, and $\mathcal{E}_s=\{e_1,e_2,...,e_m\}$ is the entity set of all $m$ entities in the sentence. In order to perform fine-grained n-ary RE, we need first to encode a span-tuple ($e_1,e_2,e_3$) consisting of every arrangement of three ordered entities, where $e_1,e_2,e_3\in \mathcal{E}_s$. The main computational consumption of Text2NKG is selecting every span-tuple of three ordered entities to encode them and get the classified labels in multiple-label classification part. If we adopt an traversal approach with each span-tuple in one training items, the time complexity will be O($m^3$).
>
> To reduce the high time complexity of training every span-tuple as one training item, Text2NKG uses packed levitated markers that pack one training item with each entity in $\mathcal{E}_s$ separately. Every input sentence contains $m$ training items with span-tuples for any ordered arrangement of three entities for multiple-label classification. Therefore, the time complexity decreased from O($m^3$) to O($m$).
>
> ---
> **For Weakness 2 (More detailed analysis and discussion on the impact of each component on different types of NKG schemas would provide deeper insights into their contributions.):**
>
> Thanks for your suggestions. In the setup of the ablation experiments, we separately removed the three main components of Text2NKG: data augmentation (DA), the null-label weight hyperparameter (α), and hetero-ordered merging (HM). DA and α are explicitly discussed in **Section 4.3 (Training Strategy)** and are part of the Multi-label Classification in **Figure 3**, representing two main training strategies. HM is part of the Hetero-ordered Merging in **Figure 3**. The ablation experiment results show that all three components contribute to the final outcome, validating the rational design of these modules.
>
> Specifically, in the **Text2NKG w/o DA** setting, we did not perform augmented training with all 6 permutations of span-tuples, only training with the original order. This led to uneven training samples, causing some relational representations to be under-trained and reducing effectiveness. In the **Text2NKG w/o α** setting, we set α to 1.0, giving the null-label the same weight as other labels. Since the null-label appears far more frequently than other labels in training, this caused the model to favor classifying as null-label, resulting in fewer extracted n-ary relational facts and a drop in recall, thus decreasing performance. In the **Text2NKG w/o HM** setting, we didn't combine the probability values of different permutations during the decoding phase, preventing the model from fully considering different ordering, leading to a decrease in performance.
>
> ---
> **For Question 1 (Can you provide an example of extracting N-ary relations in real-world information extraction?):**
>
> Take medical diagnostic decision-making applications as an example. Medical knowledge is more complex than general domain facts, with a higher proportion of n-ary relational facts. The medical fact **"A male hypertensive patient is diagnosed with mild creatinine elevation when serum creatinine is between 115-133μmol/L"** is represented as the main triplet with gender and two creatinine value indicators as auxiliary keys: **(Hypertensive patient, diagnosis, mild increase in creatinine | gender: male, numerical indicator: blood creatinine (μmol/L) ≥ 115, numerical indicator: blood creatinine (μmol/L) ≤ 133)**, forming a hyper-relational fact with five entities, more completely representing complex hypertension medical knowledge.
>
> With the Text2NKG framework, we can more easily extract NKG in hyper-relational schema from medical guidelines to model medication and treatment rules, a process previously often entirely manual. With Text2NKG's help, this process can be simplified to automated extraction and manual review. The extracted n-ary relational facts comprising the NKG can be used for subsequent tasks through link prediction, hierarchical graph, multi-hop logical queries, etc., realizing the practical application of NKG.
>
> ---
> **For Question 2 (How should NKG be stored and utilized, and can n-ary extraction be applied in RAG or Agent in the era of LLMs?):**
>
> NKGs should be stored in structured and efficient formats to facilitate easy retrieval and manipulation. Graph databases like Neo4j and Tugraph are ideal for managing n-ary relations due to their optimization for graph data. This n-ary relational fact can be represented using a special node to denote the n-ary relation, and then stored using traditional binary relational RDF. Efficient indexing and querying mechanisms, including SPARQL for RDF-based graphs, are crucial for quick data access.
>
> N-ary extraction can significantly benefit RAG models and intelligent agents in the era of LLMs. For RAG models, n-ary extraction enhances contextual understanding and enables dynamic content generation by providing richer contexts and retrieving relevant information from knowledge bases, improving GraphRAG. In LLM agents, n-ary extraction improves decision-making and interaction quality by enabling better comprehension of complex relationships and contextual reasoning. The scalability of n-ary extraction allows agents to adapt to various domains, offering versatility through custom n-ary relation schemas.

---

> > ### Comment · Reviewer_ED7F · 2024-08-14
> >
> > Thank you for the detailed rebuttal and clarifications. Your explanation of the time complexity reduction using packed levitated markers and the additional details on the ablation studies provide a clearer understanding of the computational efficiency and the contributions of each component within the framework.
> >
> > The real-world example of extracting n-ary relations in the medical domain helps illustrate the practical application of your approach. Additionally, your discussion on the storage and utilization of NKGs, particularly in relation to RAG models and intelligent agents, offers valuable insights into the potential use cases of your work.
> >
> > Overall, your rebuttal effectively addresses the concerns raised, and I appreciate the additional context provided. I remain positive about the contributions of your paper.

---

### Official Review · Reviewer_rUhA · 2024-07-11

**Soundness:** 2
**Presentation:** 2
**Contribution:** 1
**Rating:** 4
**Confidence:** 4

**Summary:**

The paper presents Text2NKG, a new framework designed for fine-grained n-ary relation extraction aimed at constructing n-ary relational knowledge graphs (NKGs). By introducing a span-tuple classification method combined with hetero-ordered merging and output merging strategies, it achieves extraction across varying entity arities while preserving their order, enhancing the granularity of NKG construction. Moreover, Text2NKG is adaptable to four prevalent NKG schemas.

**Strengths:**

**Validated Effectiveness**: Experimental results demonstrate the capability of Text2NKG in n-ary relation extraction.

**Relevance and Significance**: This work focuses on a related challenge in knowledge graph construction. Its consistency with practical application requirements enhances the importance of research.

**Weaknesses:**

**Lack of Implementation Detail**: Vital specifics regarding input prompts to language models like GPT-4, and whether the Generative Baseline and Pipelinge Baseline underwent supervised fine-tuning, are omitted. These details are crucial for reproducibility and assessing the thoroughness of comparative analysis.

**Inherent Limitations of BERT-style Architectures**: The reliance on BERT-based architectures may restrict the handling of long texts. This could limit the scalability and applicability of Text2NKG in domains with extensive contextual requirements.

**Limited application scope**: Although Text2NKG supports 4 typical NKG schemas, the framework seems to be specifically designed for these specific structures, which may limit its generality in practical applications.

**Questions:**

Please refer to the Weaknesses

**Limitations:**

The authors have addressed the limitations.

---

> ### Author Rebuttal · Authors · 2024-08-02
>
> Thank you for a detailed review. We appreciate that you find our work effective for n-ary relation extraction and consist with practical applications. We answer your concerns below.
>
> ---
> **For Weakness 1(i) (Vital specifics regarding input prompts to language models like GPT-4.):**
>
> Our ChatGPT and GPT-4 prompts both adopted a 1-shot unsupervised design. The specific input prompts (which we will add to the Appendix of the final paper) are as follows:
>
> ```
> Task: Based on the relation_list and qualifier_list and the given input sentence and entity_list, output n-ary relational facts in hyper-relational schema.
>
> relation_list=['adjacent station', 'award received', 'candidacy in election', 'capital of', 'cast member', 'chairperson', 'child', 'coach of sports team', 'connecting line', 'country', 'country of citizenship', 'director / manager', 'educated at', 'employer', 'followed by', 'head of government', 'head of state', 'headquarters location', 'home venue', 'incarnation of', 'instance of', 'league', 'legislative body', 'located in the administrative territorial entity', 'located on street', 'location', 'manufacturer', 'member of', 'member of political party', 'member of sports team', 'military branch', 'narrative role', 'noble title', 'nominated for', 'notable work', 'occupant', 'occupation', 'operator', 'original broadcaster', 'owned by', 'parent organization', 'part of', 'part of the series', 'participant', 'participating team', 'partner in business or sport', 'performer', 'place of birth', 'position held', 'present in work', 'replaces', 'residence', 'shares border with', 'significant event', 'sport', 'sports season of league or competition', 'spouse', 'stock exchange', 'subclass of', 'used by', 'voice actor', 'winner']
>
> qualifier_list=['academic degree', 'academic major', 'adjacent station', 'affiliation', 'applies to part', 'character role', 'connecting line', 'country', 'diocese', 'electoral district', 'end time', 'follows', 'for work', 'has part', 'instance of', 'located in the administrative territorial entity', 'location', 'member of political party', 'mother', 'national team appearances', 'nominee', 'number of matches played/races/starts', 'number of points/goals/set scored', 'object has role', 'of', 'performer', 'point in time', 'position held', 'position played on team / speciality', 'publication date', 'quantity', 'ranking', 'replaces', 'series ordinal', 'sports league level', 'start time', 'statement disputed by', 'statement is subject of', 'street number', 'subject has role', 'ticker symbol', 'together with', 'towards', 'winner']
>
> Example:
> Input:
> sentence='The current Leader of the National Party in the Parliament of Australia is Barnaby Joyce , and Deputy Leader is Fiona Nash , both elected on 11 February 2016 following the retirement of Warren Truss as Leader .'
> entity_list=['National Party', 'Barnaby Joyce', '11 February 2016', 'Warren Truss']
> Output:
> [(entity1='National Party', relation='chairperson', entity2='Warren Truss', qualifier1='end time', entity3='11 February 2016'), (entity1='National Party', relation='chairperson', entity2='Barnaby Joyce', qualifier1='start time', entity3='11 February 2016')]
>
> Then:
> Input:
> sentence='Apple was founded by Steve Jobs , Steve Wozniak , and Ronald Wayne on April 1 , 1976 , to develop and sell personal computers .' entity_list=['Apple', 'Steve Jobs', 'April 1 , 1976']
> Output:
> ```
>
> **For Weakness 1(ii) (Whether the Generative Baseline and Pipeline Baseline underwent supervised fine-tuning, are omitted.):**
>
> As **Section 5.1** and **Appendix D** show, we adopted the same baselines as in the CubeRE paper to compare with Text2NKG. The settings and results of the Generative Baseline and Pipeline Baseline are both from that paper. Similar to CubeRE, Text2Event, UIE, and LasUIE, the Generative Baseline and Pipeline Baseline used supervised training methods. Additionally, as shown in **Table 2**, unsupervised and supervised methods are obviously distinguished.
>
> ---
> **For Weakness 2 (The reliance on BERT-based architectures may restrict the handling of long texts.):**
>
> In **Appendix G.3**, we have analyzed how Text2NKG can address long contexts with relations spread across various sentences. If the text to be extracted is a lengthy piece, it can undergo long-form n-ary relation extraction. The maximum text segment size for our proposed method depends on the maximum text length that a transformer-based encoder can accept, such as Bert-base and Bert-large, which have a maximum limit of 512. To extract from larger documents, we simply need to switch to encoders with larger context lengths, which all serve as the encoder portion of Text2NKG and are entirely decoupled from the n-ary relation extraction technique we propose. This is one of the advantages of Text2NKG. Its primary focus is to address the order and combination issues of multi-ary relationships. We can seamlessly combine a transformer encoder that supports long texts with Span-tuple Multi-label Classification to process n-ary relation extraction in long chapters.
>
> ---
> **For Weakness 3 (Although Text2NKG supports 4 typical NKG schemas, the framework seems to be specifically designed for these specific structures.):**
>
> Text2NKG's multi-label classification strategy and hetero-ordered merging strategy allow users to freely replace custom n-ary relation schemas. When users define an n-ary relational fact composed of n entities and m relations, they only need to set the corresponding number of FNNs for the output of m categories. Then, by using user-defined rules for hetero-ordered merging, custom fine-grained n-ary relation extraction can be quickly and flexibly implemented.
>
> Plus, the hyper-relational schema, event-based schema, role-based schema, and hypergraph-based schema cover nearly all schemas of n-ary relational knowledge graphs (NKGs). In **Section 2.1**, we provide a detailed summary of existing instances of NKGs.

---

### Author Rebuttal · Authors · 2024-08-02

We thank all four reviewers for carefully reading our paper and providing constructive feedback. We appreciate the recognition of our work's strengths, which are summarized as follows.

**1. Novelty in Motivation.**
To address challenges such as the diversity of NKG schemas, determination of the order of entities, and variability of the arity of n-ary RE, we propose a novel fine-grained n-ary relation extraction framework, Text2NKG, for NKG construction. (Supported by Reviewers: ED7F: Strength 1, GMbk: Strength 1, XsBb: Strength 1)

**2. Practicality and Flexibility.**
Over 30% of real-world facts contain more than two entities, with multiple n-ary schemas. Text2NKG supports any schema of n-ary relation extraction flexibly. (Supported by Reviewers: rUhA: Strength 2, ED7F: Strength 2, GMbk: Strength 4, XsBb: Strength 3)

**3. SOTA Performance and Effectiveness.** Experimental results show that Text2NKG achieves state-of-the-art performance in fine-grained n-ary relation extraction tasks, showing its effectiveness. (Supported by Reviewers: rUhA: Strength 1, ED7F: Strength 3, GMbk: Strength 2)

**4. Extensive Verification and Analysis.**
We conducted extensive experiments to demonstrate the effectiveness of Text2NKG and each of its modules, and compare it further with advanced LLMs like GPT. We also analyzed the advantages and disadvantages of large and small models for this task, and considerations for long texts. (Supported by Reviewers: ED7F: Strength 4, GMbk: Strength 3, XsBb: Strength 2)

We believe we have addressed all points mentioned and substantially improved the paper. We responded to the four reviewers, respectively.

---

### Decision · Program_Chairs · 2024-09-25

**Decision:**

Accept (poster)

**Comment:**

The paper focuses on the extraction of n-ary relations from text and proposes a method based on a linear multi-label classification of all possible entity encoding $3$-combinations to build $3$-ary facts. They also use data augmentation (inverse relations, reordering of hyperrelational facts) and use maximum aggregation to obtain a prediction. The $3$-ary facts are combined via a fixed rule for common $(n-1)$-ary prefixes.
The paper's strengths include (1) the motivation and relevance of the task (`rUhA`, `XsBb`), (2) the writing (`ED7F`, `XsBb`), (3) the experimental results (`rUhA`, `ED7F`, `GMbk`), and (4) the number of baselines and ablations considered (`ED7F`, `GMbk`, `XsBb`).
The following shortcomings are present: (1) the scalability of the approach (`ED7F`), (2) a lack of details regarding the use of LLM baselines (`rUhA`, `GMbk`, `XsBb`; these were added during the rebuttal), (3) a lack of detailed analysis of the individual components presented, and (4) a heuristic rule-based aggregation to higher arity.
The reviewers and authors generally participated in the discussions, and some issues were resolved during the discussions.